# Pulsed Microwave Liver Ablation: An Additional Tool to Treat Hepatocellular Carcinoma

**DOI:** 10.3390/cancers14030748

**Published:** 2022-01-31

**Authors:** Giacomo Zanus, Giovanni Tagliente, Serena Rossi, Alessandro Bonis, Mattia Zambon, Michele Scopelliti, Marco Brizzolari, Ugo Grossi, Maurizio Romano, Michele Finotti

**Affiliations:** 14th Surgery Unit, Regional Hospital Treviso, University of Padua, DISCOG, 31100 Padua, Italy; giacomo.zanus@aulss2.veneto.it (G.Z.); giovanni.tagliente@aulss2.veneto.it (G.T.); serena1.rossi@aulss2.veneto.it (S.R.); alessandro.bonis@studenti.unipd.it (A.B.); mattia.zambon@studenti.unipd.it (M.Z.); michele.scopelliti@aulss2.veneto.it (M.S.); marco.brizzolari@aulss2.veneto.it (M.B.); ugo.grossi@aulss2.veneto.it (U.G.); maurizio.romano@aulss2.veneto.it (M.R.); 2Baylor Scott & White Annette C. and Harold C. Simmons Transplant Institute, Baylor University Medical Center, Dallas, TX 75204, USA

**Keywords:** liver microwave ablation, hepatocellular carcinoma, mini-invasive approach

## Abstract

**Simple Summary:**

Hepatocellular carcinoma (HCC) is the seventh most frequent neoplasm and the second most common oncologic cause of death, mostly in patients with end-stage liver disease. HCC treatment is complex and different solutions are available, ranging from liver transplants to local therapies. In this study, we analyze the role of pulsed microwave liver ablation as an additional treatment option.

**Abstract:**

This study aimed to analyze the outcomes of HCC patients treated with a novel technique—pulsed microwave ablation (MWA)—in terms of safety, local tumor progression (LTP), intrahepatic recurrence (IHR), and overall survival (OS). A total of 126 pulsed microwave procedures have been performed in our center. We included patients with mono- or multifocal HCC (BCLC 0 to D). The LTP at 12 months was 9.9%, with an IHR rate of 27.8% at one year. Survival was 92.0% at 12 months with 29.4% experiencing post-operative complications (28.6% Clavien–Dindo 1–2, 0.8% Clavien–Dindo 3–4). Stratifying patients by BCLC, we achieved BCLC 0, A, B, C, and D survival rates of 100%, 93.2%, 93.3%, 50%, and 100%, respectively, at one year, which was generally superior to or in line with the expected survival rates among patients who are started on standard treatment. The pulsed MWA technique is safe and effective. The technique can be proposed not only in patients with BCLC A staging but also in the highly selected cases of BCLC B, C, and D, confirming the importance of the concept of stage migration. This procedure, especially if performed with a minimally invasive technique (laparoscopic or percutaneous), is repeatable with a short postoperative hospital stay.

## 1. Introduction

Hepatocellular carcinoma (HCC) is the most common primary liver cancer, accounting for more than 80% of liver tumors, and usually arises in the context of liver cirrhosis. Chronic viral hepatitis and alcohol-related liver disease are the main risk factors for liver dysfunction and HCC [1]. However, especially in high-income countries, nonalcoholic fatty liver disease (NAFLD) is a growing risk factor for HCC [2]. HCC is correlated with low survival, even in developed countries, leading to a 5-year age-standardized survival rate of <20% [3]. However, when it is diagnosed in the early stages, treatment options can notably improve the outcome [4]. Wide spectrums of therapeutic approaches are feasible, with curative, bridging, downstaging, or palliative intent [5].

Potential curative treatments, such as liver transplantation (LT), liver resection (LR), and locoregional ablation (LRA, with microwave or radiofrequency methodology), offer, in the selected patient, the best outcome in terms of overall survival (OS) and disease-free survival (DSF). LR, LRA, and transcatheter arterial chemoembolization (TACE) are often proposed as methods for downstaging or bridging HCC patients to LT [5,6]. In the last decade, systemic therapies have undergone an important expansion based on multi-kinase inhibitors (sorafenib and regorafenib), vascular endothelial growth factor receptor inhibitors (lenvantinib and cabozantinib), or immunotherapy (nivolumab and pembrolizumab) [7]. The Barcelona Clinic Liver Cancer (BCLC) staging system is the most common algorithm used to guide treatment decisions, wherein the treatment type is selected based on prognostic variables and the stage of the disease. The BCLC was generated from the analysis of randomized controlled studies and has been endorsed by many guidelines, including AASLD, EASL-EORTC, and ESMO-ESDO [8,9,10]. In the BCLC system, LT, LR, and LRA (potential curative tools) are restricted to very early and early stages (BCLC 0 and A). The intermediate stage (BCLC B) comprises heterogeneous HCC patients, and TACE is considered the standard of care. Single large to multinodular HCC, with no macrovascular invasion or extrahepatic spread in patients with conserved liver function (Child–Pugh Score A–B), is covered in this stage. Palliative treatments are also proposed in stages C and D with medical therapy and best supportive care, respectively. However, many authors considered the BCLC algorithm too strict, especially in stage B [11,12].

The gold standard therapy for patients with BCLC 0–A remains resection [13]. LRA, especially laparoscopic MWA, is considered an important tool with good outcomes among patients [8,11,14,15,16,17]. 

Pulsed microwave ablation is an evolution of the original microwave ablation technique. This approach has been used in murine models or ex vivo experiments, but no data exist on its use in treating HCC. The pulsed mode is characterized by the succession of supply and pause cycles. Thanks to the efficient cooling of the antenna stem during the supply pause, the temperature in the tissues closest to the antenna decrease faster than those in the periphery; therefore, with a pulsed algorithm, it is reasonable to expect a significant increase in the sphericity index to minimize the amount of healthy tissue involuntarily involved and to preserve critical structures close to the target. This can help overcome the heat-sink effect, creating larger ablation zones than when using lower, continuous power. To date, to the best of our knowledge, no clinical series on pulsed MWA for the treatment of HCC are present in the literature. This study aims to evaluate the application and outcomes of pulsed microwave ablation in the treatment of HCC in the context of liver cirrhosis at different stages of the BCLC.

## 2. Material and Methods

We retrospectively collected the data of patients treated with pulsed MWA at a tertiary hepatobiliary center for liver and pancreatic surgery (Treviso Hospital, Italy) from a prospective database between October 2018 and February 2020.

The primary endpoint was the safety and effectiveness of this novel procedure, considering postoperative complications according to the Clavien–Dindo classification [18].

The secondary endpoint was to evaluate local tumor progression (LTP), intrahepatic recurrence (IHR), and overall survival (OS).

The inclusion criteria were:−Patients not suitable for liver resection (due to performance status, portal hypertension, or the status of end-stage liver disease)−The absence of extrahepatic spread or vascular invasion−HCC < 70 mm

We obtained clinical and laboratory parameters to calculate MELD scores and Child–Pugh scores to verify the burden of disease of our patients. We considered earlier treatments, ablation parameters, post-operative complications (Clavien–Dindo), and the duration of hospital stay.

### 2.1. Pulsed Microwave Ablation

Pulsed microwave ablations were performed with the AMICA PROBE, AMICA gen AGN H 1.2. Intraoperative USs were performed using a flexible, linear ALOCA UST-5550 33 mm probe at a frequency of 5–10 MHz. The surgical approach was described previously [19,20].

MW ablation is considered pulsed when, during the time of the procedure, we can identify a period called T-on, where the ablation is delivered to the tissue, and a time called T-off, where tissue does not receive any ablation. This method may improve the sphericity of the ablation zone in contrast with the continuous technique [21].

### 2.2. Radiological Follow-Up

Patients were followed-up with at 1, 3, 6, 9, and 12 post-operative months with labs, AFP, and a CT abdominal scan or MRI with contrast.

LTP was defined as a lesion born in the same place where the ablation was performed, with evidence of radiological enhancement, which was not visible before surgery. IHR was defined as the radiological evidence of a new lesion within the liver [22].

Two independent radiologists reviewed the images.

### 2.3. Statistical Analysis

We analyzed the data using IBM SPSS statistics v.2.6, considering OS, DFS, LTP, and IHR. We considered statistical significance to be *p* < 0.05. Continuous variables are represented as medians (ranges). Categorical or nominal variables are presented as frequencies (%). We used the Wilcoxon test and Student’s *t*-test to compare quantitative variables; we tested categorical variables using the chi-squared test. Overall survival and disease-free survival rates were calculated using the log-rank test and Kaplan–Meier analyses. Then, statistics were used to draw estimates of the survival curves considering two different parameters: nodules’ size (<30 mm, between 30 and 50 mm, and >50 mm) and the BCLC stage before surgery.

## 3. Results

Between October 2018 and February 2020, during the treatment of 286 HCC nodules in 113 patients, 126 procedures were performed. The characteristics of patients are represented below (Table 1).

### 3.1. Population

We treated 88 males (77.9%) and 25 females. Viral etiology was the most common (60%), and HCV was the most common viral agent (38%).

In our sample, 60% were BCLC 0–A (*n* = 11 BCLC 0, *n* = 59 BCLC A), 30% were BCLC B (*n* = 33), 5% were BCLC C (*n* = 6) and 2% were BCLC D (*n* = 2). The median MELD score was 9.4 ± 3.1, and the Child–Pugh score was A in 81.4% of cases (40 cases were A4, 52 cases were A5). A total of 92 cases had Child–Pugh score of A, and a Child–Pugh score of B was assigned for 14.2% (*n* = 16) of cases (Table 1 and Table 2).

### 3.2. Disease

A total of 90 (79%) of our patients presented with multinodular disease, and 60% of them were treated in a multimodal fashion. Further, 85 patients (67.5%) out of our sample presented with portal hypertension and 33 (26.4%) presented with ascites during preliminary surgical exploration. Radiological evidence of portal thrombosis was clear in 10 cases (8.1%). Pre-operative CEA, CA 19.9, and AFP values were, respectively, 3.4 ± 4.9 ng/mL, 28.7 ± 37.3 U/mL, and 51.0 ± 138.1 ng/mL (Table 2).

Overall, 253 nodules (88.5%) of HCC were smaller than 30 mm, 29 (10%) were from 30 to 50 mm, and just 4 (1.4%) were bigger than 50 mm. Further, 132 (47.2%) nodules were in a back segment. Patients had a mean of 3.5 ± 2.2 nodules (range 1–10), with a median of 3 lesions (Table 3).

### 3.3. Procedures

We performed 126 procedures (105 VLS, 14 open, and 7 percutaneous). The median time of ablation was 8.0 ± 7.0 min. The median ablation power was 81.2 ± 19.7 W. In 18.6% of procedures, the pulsed MWA was associated with synchronous resection (23 cases of minor resection and one major resection). One-third of the procedures needed to be completed with another MW; TACE or TARE was performed in another 34% of cases (Table 3).

### 3.4. Complications According to Clavien–Dindo and Outcome

In 126 procedures, we registered 16 cases of decompensated ascites (12.7%), 13 cases of fever (6 cases of post-procedural fever (4.7%) and 7 cases of infection (5.6%)), 5 cases of anemia (4%), and 3 cases of hepatic encephalopathy (2.4%).

According to the Clavien–Dindo score, 28.6% of them were Clavien–Dindo 1 or 2; we registered one case of Clavien–Dindo 4 (the patient was sent to the ICU after surgery, just for one day).

The general median hospital stay was 4.96 days (range 1–26), with ranges of 8.15, 4.84, and 1 day for open, VLS, and percutaneous approaches, respectively. No intra- or perioperative mortalities (across 30 days) were reported.

Nine patients died during our 12-month follow-up period (8%), and only one patient died before three months. The survival rate was 92.0% at the 12-month follow-up. When stratifying patients by BCLC stage, we achieved respective BCLC 0, A, B, C, and D survival rates of 100%, 93.2%, 93.3%, 50%, and 100% at one year. These rates are generally superior to or in line with the expected survival rates for patients who are started on standard treatment, respectively: 90% at 5 years, median 36 months, median 16 months, median 6–8 months, and 3–4 months. Considering the BCLC stage, the product-limit survival estimates are represented in Figure 1.

### 3.5. Follow-Up

Overall, 9.9% of HCC recurred at 12 months (8.4% for nodules smaller than 30 mm, 21.1% between 30 and 50 mm, and 25% for lesions bigger than 50 mm).

Considering BCLC stages, we obtained, respectively, a 12-month LTP of 9.1% (0), 16.9% (A), 3.9% (B), and 50% (C and D), with statistical significance among groups (*p* = 0.01) (Table 4).

The IHR was 27.8%, with no statistical significance when considering the diameter of the nodules among groups.

## 4. Discussion

For the first time, we report a case series of HCC patients treated with pulsed microwave ablation. Scarce experiences of pulsed MW ablation have been reported in animal series [21,23,24].

Radosevic et al. recently reported a computer model to compare pulsed vs. continued microwave ablation, suggesting that the pulsed mode may create better sphericity [25]. To date, a debate is ongoing as to whether pulsed or continuous ablation is performing better. Pulsed microwave ablation, with its intermittent energy administration, seems to be able to deliver high power across a specific time frame, considering that the peak of the power is strictly correlated with the area of ablation [21,23,26,27]. However, the main limitation of these studies is that the ablation is applied to the non-cirrhotic liver, where the energy dispersion is completely different compared to a cirrhotic liver. We believe that, in fibrotic tissue, pulsed energy is able to achieve a more uniform temperature profile compared to the continuous method, due to the different types of blood perfusion and coagulation tissues seen in the cirrhotic liver. Further studies with the pathological evaluation of this population are needed.

The novelty of the study, other than the use of the pulsed technique on cirrhotic patients with different surgical approaches, is the selection strategy of patients who underwent pulsed MWA. The BCLC algorithm was adopted many years ago, thanks to the ease of interpretation and application of the algorithm: each stage is associated with a clear and direct treatment option. However, over the last years, critical limitations have been raised, mainly related to the prognostic capability but also its treatment options. With the advancements in HCC management and therapeutic options, many authors consider the BCLC classification and its treatment algorithm too rigid and obsolete.

The new upgrade of the BCLC 2022 includes a discussion about treatments considering two aspects: the treatment stage migration (TSM) and the untreatable progression (UTP) [28,29,30]. Nowadays a modern, multidisciplinary, and individual approach to the patient is mandatory, as confirmed by our series, especially within the BCLC B stage that continues to include a heterogeneous group of patients [31].

Our institution used wide inclusion criteria, ranging from BCLC 0 to BCLC D (60% of the patients were BCLC 0-A, 30% were BCLC B, 5% were BCLC C, and 2% were BCLC D). Furthermore, most of the patients presented with multinodular disease, with 8.1% presenting with portal thrombosis and 26% presenting with clinically significant abdominal ascites; 60% of these patients were previously treated with other treatments (previous liver resection, other MWA/RFA, TACE/TARE, or PEI). This population reflects the “real life” of clinical practice, often far from the BCLC indications, and must be considered during the evaluation of the study results.

### 4.1. Safety and Efficacy

Across 126 procedures, most of the complications were Clavien–Dindo 1 or 2 (28.6%), related to decompensated ascites (12.7%), fever (4.7%; compatible with the MWA procedure), anemia (4%), and hepatic encephalopathy (2.4%). One patient was sent to the intensive care unit because he developed respiratory failure, reflecting the frailty of our population. To note, 37% of the patients were BCLC staged B, C, or D before the procedure and 132 (47.2%) nodules were in posterior segments.

Despite this, the complication rates are comparable to other series [32,33,34,35], with a slightly longer general median hospital stay of 4.96 days (range 1–26). Curiously, a multicenter study reported just 2.9% of complications were Clavien–Dindo 1 or 2, but among 14 Centers in Italy, only 4 of them used videolaparoscopy (which we used currently), with a preferred percutaneous approach [36]. This reflects the high-risk profile of the patient, as discussed above. Moreover, updates in surgical skills were recently reported for approaching multi-treated patients with lesions located in difficult sites [15], but we only used laparoscopic access to treat our cohort. However, no intrahospital mortality has been recorded compared to the 0.4% in-hospital mortality rate described in series with similar pre-operative high-risk patient selection (high MELD scores and BCLC stages B–C) [32], pointing out the feasibility of the procedure in this population.

Considering the BCLC stage, we obtained, respectively, at the 12-month follow-up, an LTP of 9.1% for BCLC 0 patients, 16.9% for BCLC A patients, 3.9% for BCLC B patients, and 50% for BCLC C and D patients, with statistical significance among the groups (*p* = 0.01). The LTP among the BCLC 0 and A groups are comparable with other series (as reported in Table 5, and MWA efficacy is strictly related to the HCC dimension, with an LTP, respectively, of 8.4%, 21.1%, and 25% for tumors sized <3 cm, 3–5 cm, and >5 cm, in our experience. The pulsed technique allows for the delivery of high-power ablation, reducing the effect of heat dispersion. Moreover, it does not increase the risk of harmful vascular phenomena, thanks to the high-flow cooling antenna and better target precision allowed by the sphericity.

### 4.2. Survival

According to EASL guidelines, the expected survival rate is 90% for patients who are BCLC 0 and 50–70% for patients who are BCLC A at 5 years. Vitale et al. approximately described, for BCLC 0–A, an expected OS of 87% and 85%, respectively [44]. In the literature, OS at one year is between 80% and 98% [22,32,33,37,41,42]. In our center, we observed a one-year survival rate of 100% for patients who were BCLC 0 and 93.2% for patients who were BCLC A.

These results, as reported by other series, reflected the importance of a multidisciplinary evaluation, and good outcomes can be obtained by avoiding the too-strict criteria related to the BCLC algorithm. An alternative approach to the HCC treatment is described by the ITA.LI.CA staging system for treatment allocation. In this allocation system, the treatment is not determined by the BCLC stage (stage hierarchy), but by the most effective therapy feasible for that patient (therapeutic hierarchy) [45,46,47].

Considering BCLC D patients, no death was reported at one year, despite the burden of liver disease, but these data must be interpreted while considering the small sample of BCLC D patients involved (2%).

## 5. Conclusions

In conclusion, for the first time, we showed that the pulsed MWA technique is safe and effective for the treatment of HCC in cirrhotic patients. We proposed that pulsed microwaves could be applied even in a high-risk population in terms of frailty and tumor burden, to the detriment of a slightly longer hospital stay but with excellent results in terms of safety and tumor control. Furthermore, the use of an aggressive surgical approach during the intermediate stage (BCLC B) can downstage the disease, allowing the patient to be evaluated for a possible liver transplant.

## Figures and Tables

**Figure 1 cancers-14-00748-f001:**
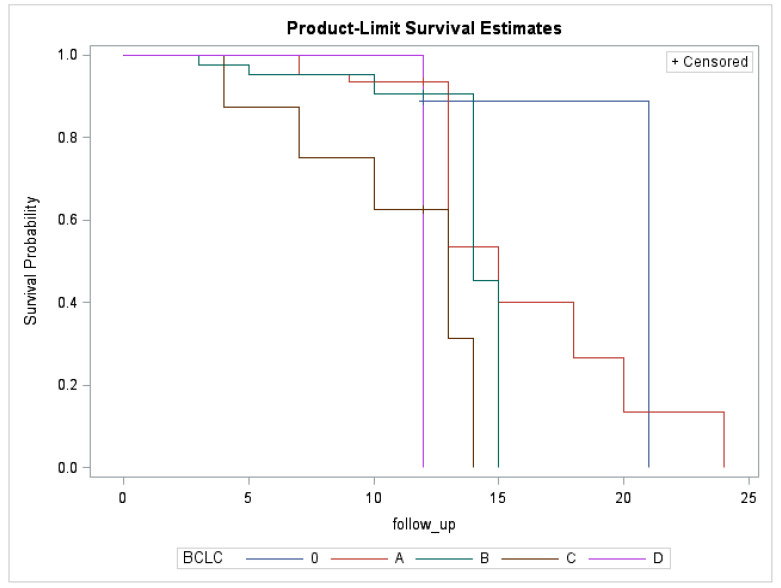
Survival rates of patients according to BCLC staging.

**Table 1 cancers-14-00748-t001:** Population parameters.

Variables	Results
Age, mean ± SD (range); median	68.2 ± 9.2 (36–88); 69
Sex (M/F), *n* (%)	88 (77.9)/25 (22.1)
Etiology, *n* (%)	
Alcohol-related	22/113 (19.5)
Alcohol-related and metabolic	5/113 (4.4)
Viral	68/113 (60.2)
HCV	43/113 (38.1)
HCV + HIV	2/113 (1.8)
HCV + ETOH	2/113 (1.8)
HBV	17/113 (15.0)
HBV + ETOH	4/113 (3.5)
Metabolic	11/113 (9.7)
Other	7/113 (6.2)
BCLC, *n* (%)	
0	11 (10)
A	59 (52)
B	33 (29)
C	6 (5)
D	2 (2)
NR	2 (2)

**Table 2 cancers-14-00748-t002:** Disease parameters in our sample.

MELD, Mean ± SD (Range)	Results
Considering procedures	9.4 ± 3.1 (6–21)
Considering patients (MELD score at the first procedure)	9.4 ± 3.2 (6–21)
Child–Pugh, *n* (%)	
A5	52 (46.0)
A6	40 (35.4)
B7	13 (11.5)
B8	2 (1.8)
B9	1 (0.9)
C10	2 (1.8)
C11	1 (0.9)
*NR = not reported/missing data	2 (1.8)
Portal thrombosis, *n* (%)	10 (8.1)
CEA (ng/mL), mean ± SD (range); median	3.4 ± 4.9 (0.5–49.4); 2.3
CA 19.9 (U/mL), mean ± SD (range); median	28.7 ± 37.3 (0.6–296); 18
AFP (ng/mL), mean ± SD (range)	51.0 ± 138.1 (1–851)
Ascites, *n* (%)	33 (26.4)
Portal hypertension, *n* (%)	83 (67.5)
*NR = not reported/missing data	

**Table 3 cancers-14-00748-t003:** Surgical and technical data.

Lesions’ size (mm), mean ± SD (range); median	17.3 ± 11.7 (7–70); 13
Clusters of diameters, *n* (%)	
<3 cm	253/286 (88.5)
3–5 cm	29/286 (10.1)
>5 cm	4/286 (1.4)
Site of lesions, *n* (%)	
Posterior segments (s1, s6, s7, s8)	132 (47.2)
Anterior segments and left lobe (s2, s3, s4a, s4b, s5)	154 (53.8)
Multinodular disease, *n* (%)	90 (78.9)
Number of lesions in patients, mean ± SD (range); median	3.5 ± 2.2 (1–10); 3
Previous surgical treatments (*n*°), mean ± SD (range); median	1.9 ± 2.2 (0–11); 3
Previous loco-regional treatments (*n*°), mean ± SD (range)	
PEI	0.1 ± 0.7 (0–6)
RFA/MWA	0.5 ± 0.8 (0–4)
TACE	0.7 ± 0.9 (0–4)
Previous resection (*n*°), mean ± SD (range); median	0.2 ± 0.5 (0–2)
Surgical approach, *n* (%)	
VLS	105 (83.3)
Laparotomic	14 (11.1)
Percutaneous	7 (5.6)
Time of ablation (min), mean ± SD (range); median	8.0 ± 7.0 (1–40); 6.0
Probe power (W), mean ± SD (range); median	81.2 ± 19.7 (40–120); 75
Associated resection, *n* (%)	24 (18.6)
Major	1 (0.8)
Minor	23 (18.0)
Hospital stay (days), mean ± SD (range); median	
All	4.96 (1–26); 4
VLS (105)	4.84 (1–26); 4
Open (14)	8.15 (4–20); 7
Percutaneous (7)	1.0 (1); 1
Hospital stay for patients scored Clavien–Dindo 1, 2, and 3	
*n* = 41 (32.5%)	5.09 (1–26); 4

**Table 4 cancers-14-00748-t004:** Local tumor progression (LTP) and intrahepatic recurrence (IHR) of disease.

12-Month LTP	*n* = 286
0	182 (90.1%)
1	20 (9.9%)
12-month LTP according to nodule size (mm)	*n* = 286—*p* = 0.0956
<30 (*n* = 253)	15 (8.4%)
30–50 (*n* = 29)	4 (21.1%)
>50 (*n* = 4)	1 (25%)
12-month LTP according to the BCLC stage	*n* = 286—*p* = 0.0111
0 (*n* = 13)	1 (9.1%)
A (*n* = 113)	14 (16.9%)
B (*n* = 137)	4 (3.9%)
C (*n* = 15)	0 (0.0%)
D (*n* = 8)	1 (50.0%)
12-month IHR	*n* = 126
0	70 (72.2%)
1	27 (27.8%)
12-month IHR according to nodule size (mm)	*n* = 126—*p* = 0.7636
<30 (*n* = 103)	22 (27.8%)
30–50 (*n* = 21)	4 (25.0%)
>50 (*n* = 2)	1 (50.0%)

**Table 5 cancers-14-00748-t005:** Comparison to other experiences in the literature.

Author, Year	Procedures (*n*°)	Diameter (cm)	Follow-Up (months)	LTP (%)
Lu MD et al., 2005 [37]	49	≤3 cm 3–5 cm	25.1	6.8% 30.0%
Ohmoto K et al., 2009 [38]	49	≤2 cm	12	13%
Ding J et al., 2013 [39]	113	≤3 cm 3–5 cm	18.3	7.3% 21.2%
Vogl TJ et al., 2015 [40]	28	≤5 cm	12	11.1%
Li W et al., 2017 [41]	60	≤3 cm	12	14.9%
Xu Y et al., 2017 [42]	294	≤3 cm 3–5 cm 5–6 cm	12	10.6% 16.9% 28.6%
Baker EH et al., 2017 [22]	219	1–6 cm (median 3.2 cm)	9.9	8.5%
Santambrogio et al., 2017 [43]	60	≤3 cm	26.9	8.3%
Liu W et al., 2018 [41]	126	≤3 cm	36.8	11.7%
Cillo U et al., 2019 [32]	815	≤5 cm	6	23.1%
Our experience—2020	126	All <3 cm 3–5 cm >5 cm	12	9.9% 8.4% 21.1% 25%

## Data Availability

Data are contained within the article.

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
