# Peer review of "Pulsed Microwave Liver Ablation: An Additional Tool to Treat Hepatocellular Carcinoma"

_cancers, 2022, doi:10.3390/cancers14030748_

Round 1

Reviewer 1 Report

Abstract- edit- 'in our centre" or "at our centre" instead of to our centre

Results- The heading 3.3 "Surgery" is confusing- it could be replaced with "Ablation" or "Procedure" or Pulsed ablation. As not all lesions were ablated at Open or Laparoscopic surgery.

Is it also possible to give any data related to cost difference (was there any difference in costs for pulsed ablation vs continuous mode?). Also do you have any comparative data historical or otherwise, of continuous mode ablation to compare with pulsed ablation?

Discussion

As you mentioned BCLC is outdated and obsolete, in terms of it's treatment algorithms- do you use any other staging system in conjunction with it? BCLC certainly is very conservative in the treatment algorithm and pulsed microwave ablation is one of the modalities which certainly  looks promising for the Intermediate group (as shown by your paper)

Author Response

Comments and Suggestions for Authors

Abstract- edit- 'in our centre" or "at our center" instead of to our center

Reply: thank you for your comment. We changed to in our center

Results- The heading 3.3 "Surgery" is confusing- it could be replaced with "Ablation" or "Procedure" or Pulsed ablation. As not all lesions were ablated at Open or Laparoscopic surgery.

Reply: thank you for your comment. We changed “Procedures”

Is it also possible to give any data related to cost difference (was there any difference in costs for pulsed ablation vs continuous mode?). Also do you have any comparative data historical or otherwise, of continuous mode ablation to compare with pulsed ablation?

Reply: thank you for your comment. This is a very important topic. Cost-effectiveness evaluation is our next step, including a direct comparison between the continuous and the pulsed ablation. We are planning to include this information in the next study.

Discussion

As you mentioned BCLC is outdated and obsolete, in terms of it's treatment algorithms- do you use any other staging system in conjunction with it? BCLC certainly is very conservative in the treatment algorithm and pulsed microwave ablation is one of the modalities which certainly  looks promising for the Intermediate group (as shown by your paper)

Reply: thank you very much for your comment. This allows us to discuss a concept that our group is carrying on about the limitation of the BCLC. We use the concept that inspired the ITA.LI.CA staging system. We added this part to the discussion:

“…These results, as reported by other series, reflected the importance of the multidisciplinary evaluation, and good outcomes can be obtained by avoiding the too strict criteria related to the BCLC algorithm. An alternative approach to the HCC treatment is described by the ITA.LI.CA staging system for treatment allocation. In this allocation system, the treatment is not led by the BCLC stage (stage hierarchy) but by the most effective therapy feasible for that patient (therapeutic hierarchy)...”

Reviewer 2 Report

I think that this is an interesting paper about a medical unmet need about a rare but highly deadly disease.

The procedure is feasible and safe and it seems to be more effective as bridge to other treatments and for the local diseases.

I think that con be improved the graphic representation of outcomes.

Other concern is the concomitants therapies. This procedure was associated to other therapies that could affect the outcomes?

Author Response

I think that this is an interesting paper about a medical unmet need about a rare but highly deadly disease.

The procedure is feasible and safe and it seems to be more effective as bridge to other treatments and for the local diseases.

I think that con be improved the graphic representation of outcomes.

Other concern is the concomitants therapies. This procedure was associated to other therapies that could affect the outcomes?

Reply: Thank you for your comment. As we stated in the paper: “…In 18.6% of procedures, the pulsed MWA was associated with synchronous resection (23 cases of minor resection and one major resection)…” However, the analyses of the local tumor recurrence were performed by procedures/nodules, and the nodules that required additional treatment (like resection) have been considered as recurrence in the final analyses. As described by Cillo et al (10.1002/jso.25651) in one of the largest European series, it’s difficult to evaluate only the patient treated with only MWA, due to the natural history of Hepatocellular carcinoma (recurrence) and the often multiple treatments that the same patient can go through. The goal of this paper is to show that this procedure can be an additional tool to the broad therapeutic options.

Reviewer 3 Report

The above article represents quality work, moreover it brings novelty to the reader. The study is correctly designed and the conclusions are well supported by the results.

Author Response

The above article represents quality work, moreover it brings novelty to the reader. The study is correctly designed and the conclusions are well supported by the results.

Reply: thank you for your time and revision.